# Nonsense mutation suppression is enhanced by targeting different stages of the protein synthesis process

Amnon Wittenstein[1], Michal Caspi[1], Ido Rippin[2], Orna Elroy-Stein[3], Hagit Eldar-Finkelman[2], Sven Thoms[4], Rina Rosin-Arbesfeld[1]*

1 Department of Clinical Microbiology and Immunology, Faculty of Medicine, Tel Aviv University, Tel Aviv, Israel, 2 The Department of Human Molecular Genetics & Biochemistry School of Medicine, Tel Aviv University, Tel Aviv, Israel, 3 Shmunis School of Biomedicine and Cancer Research, George S. Wise Faculty of Life Sciences, Tel Aviv University, Tel Aviv, Israel, 4 Biochemistry and Molecular Medicine, Medical School EWL, Bielefeld University, Bielefeld, Germany

* arina@tauex.tau.ac.il

**Data Availability Statement:** All relevant data are within the paper and its Supporting Information files.

## Abstract

The introduction of premature termination codons (PTCs), as a result of splicing defects, insertions, deletions, or point mutations (also termed nonsense mutations), lead to numerous genetic diseases, ranging from rare neuro-metabolic disorders to relatively common inheritable cancer syndromes and muscular dystrophies. Over the years, a large number of studies have demonstrated that certain antibiotics and other synthetic molecules can act as PTC suppressors by inducing readthrough of nonsense mutations, thereby restoring the expression of full-length proteins. Unfortunately, most PTC readthrough-inducing agents are toxic, have limited effects, and cannot be used for therapeutic purposes. Thus, further efforts are required to improve the clinical outcome of nonsense mutation suppressors. Here, by focusing on enhancing readthrough of pathogenic nonsense mutations in the *adenomatous polyposis coli* (APC) tumor suppressor gene, we show that disturbing the protein translation initiation complex, as well as targeting other stages of the protein translation machinery, enhances both antibiotic and non-antibiotic-mediated readthrough of nonsense mutations. These findings strongly increase our understanding of the mechanisms involved in nonsense mutation readthrough and facilitate the development of novel therapeutic targets for nonsense suppression to restore protein expression from a large variety of disease-causing mutated transcripts.

## Introduction

Nonsense mutations are single nucleotide substitutions in the coding regions that result in premature termination codons (PTCs) and produce truncated, mostly nonfunctional proteins [1]. A meta-analysis based on the human gene mutation databases concluded that nonsense mutations are responsible for approximately 11% of all gene aberrations associated with inheritable diseases [2].

**Funding:** This work was supported by the The German-Israel Foundation (GIF) grant I-1459-412.13/2018 (to RRA and ST). The funders had no role in study design, decisions or publish or preparation of the manuscript.

**Competing interests:** The authors have declared that no competing interests exist.

**Abbreviations:** 4EBP-1, 4E (eIF4E)-binding protein 1; ANOVA, analysis of variance; APC, adenomatous polyposis coli; CF, cystic fibrosis; CRC, colorectal cancer; DMD, Duchenne muscular dystrophy; DMEM, Dulbecco's Modified Eagle's Medium; eEF1A, eukaryotic elongation factor 1A; eEF2, eukaryotic elongation factor 2; eEF2K, eukaryotic elongation factor 2 kinase; eIF4E, eukaryotic translation initiation factor 4E; eIF4G, eukaryotic translation initiation factor 4G; eRF1, translation termination factor 1; FCS, fetal calf serum; GM, gentamicin; MEF, mouse embryo fibroblast; MNK, MAPK interacting kinase; mTOR, mammalian target of the rapamycin; nc-tRNA, near-cognate tRNA; PTC, premature termination codon; RPMI, Roswell Park Memorial Institute; S6K1, ribosomal protein S6 kinase beta-1; Tr-APC, truncated APC; TSC, tuberous sclerosis complex; WT, wild-type.

Different compounds and small molecules can induce PTC readthrough, leading to misinterpretation of the PTC as a sense codon, thereby restoring protein synthesis (reviewed in [3]). Genetic and biochemical studies have shown that these nonsense mutation readthrough agents act by binding a specific site on the rRNA, which causes the ribosome to introduce an amino acid instead of releasing the mRNA chain. Although little is known about the exact nature of the amino acid inserted or the precise readthrough mechanism, translation through the PTC often results in the expression of a full-length protein [4]. Aminoglycoside antibiotics were the first drugs shown to induce PTC-readthrough [5], by enabling the misincorporation of near-cognate tRNA (nc-tRNA) at the A-site of the ribosome, leading to the expression of full-length proteins [6,7]. Aminoglycosides function by a mechanism that competes with translation termination [8]. However, the lack of specificity, modest readthrough effects, and toxicity of the aminoglycosides have led to the search for more efficient agents [9–14]. Additional nonsense mutation readthrough-inducing compounds that increase protein production in several cell culture and animal disease models have been identified, but the readthrough levels were usually low, achieving no more than 5% of wild-type protein expression, and most compounds have not reached the clinic [15]. Although nonsense mutation readthrough usually yields only a small percentage of the normal expression levels of the full-length protein, in some cases, such as in lysosomal storage disease, even 1% of normal protein function may restore a near-normal or clinically less severe phenotype [16,17], this threshold is disease and gene dependent as for cystic fibrosis (CF), it has been shown that 10% to 35% of CFTR activity might be needed to alleviate pulmonary morbidity significantly [18]; in Duchenne muscular dystrophy (DMD) —1% to 30% of the full-length dystrophin protein is needed [3]. It has also been demonstrated that readthrough activity can be chemically potentiated [19].

As a large number of genetic diseases result from nonsense mutations [15], identifying and developing new therapeutic strategies by better understanding the mechanism that underlines induced nonsense mutation readthrough activity is of great interest.

*Adenomatous polyposis coli* (APC) is a multifunctional tumor suppressor gene mutated in approximately 80% of sporadic and hereditary colorectal cancer (CRC) syndrome tumors [20–22]. APC inhibits the activity of the oncogenic β-catenin protein as well as functions in cell cycle control, differentiation, and apoptosis [23–25]. Mutations in APC are thought to be one of the key factors driving cancer initiation [26]. A large number of the APC mutations are nonsense mutations [27], resulting in a truncated, unfunctional protein. Here, we aimed to explore mechanisms that induce nonsense mutation suppression, leading to restored expression of the full-length APC protein.

Regulating protein synthesis is crucial for cell survival, and thus, translational dysregulation leads to aberrant growth and tumorigenicity (reviewed in [28]). In this study, we show that inhibiting translation initiation by disturbing the translation initiation complex, as well as targeting the elongation or termination stages of protein synthesis, increases induced nonsense suppression. Our results expose the complex relationship between nonsense mutation readthrough and the translation machinery and offer novel approaches to improve full-length protein production from genes containing nonsense mutations.

## Materials and methods

### Cell culture

WT and $TSC^{-/-}$ MEFs, APC 1450X cells [29], and human colon carcinoma cell lines were cultured in Dulbecco's Modified Eagle's Medium (DMEM) supplemented with 10% fetal calf serum (FCS) and 100 U/ml penicillin-streptomycin. DU4475 was cultured in Roswell Park Memorial Institute (RPMI) 1640 Medium with 10% FCS and 100 U/ml penicillin-

streptomycin. Cells were kept in a humidified 5% $CO_2$ atmosphere at 37˚C. All the following cell lines were from ATCC: COLO 320—ATCC CCL-220, SW403—ATCC CCL-230, SW620—ATCC CCL-227, SW837—ATCC CCL—235, DU4475—ATCC HTB-123, HCT116—ATCC CCL-247, and SW1417—ATCC CCL-238. The SW48 cell line was a kind gift from Prof. Uri ben David. MEF cells deficient in TSC1/2 (MEF TSC1/2$^{-/-}$) and matched MEF cells were generously provided by Dr. Kwiatkowski (Harvard Medical School, Boston, Massachusetts, United States of America) [30].

## Antibodies and reagents

The following antibodies and reagents were used: anti-GFP (mouse monoclonal; Santa Cruz; sc-9996, 1:750), anti-APC (rabbit polyclonal; Santa Cruz; sc-7930, 1:500), anti-active β-catenin (rabbit polyclonal; Cell Signaling Technology; D2U8Y, 1:2,000), anti-tubulin (mouse monoclonal; Sigma; T6199, 1:10,000), Anti-p-S6K1 (rabbit polyclonal; Cell Signaling Technology; #9205, 1:1,000), Anti-S6K1 (rabbit polyclonal; Cell Signaling Technology; #9202, 1:1,000), Anti-4EBP-1 (rabbit polyclonal; Abcam, ab2606, 1:1,000), Anti-p-S6 rabbit polyclonal; Cell Signaling Technology; #2211, 1:1,000), Anti-p-eIF4E (rabbit polyclonal; Cell Signaling Technology; #9741, 1:1,000), Anti-TSC2 (rabbit polyclonal; Cell Signaling Technology; #4308, 1:1,000), anti-mouse and anti-rabbit-HRP (Jackson Laboratories, 1:10,000), Gentamicin sulfate (Biological Industries,03–035), G418 sulfate (Mercury-ltd, CAS 108321-42-2), Rapamycin (AdooQ BioScience, A10782) in DMSO, Torin-1 (Caymanchem, 10997) in DMSO, PF-4708671 (AdooQ BioScience, A11755) in DDW, and 4EGI-1 (AdooQ BioScience, A14199) in DMSO, Tomivosertib (caymanchem, 21957) in DMSO, A-484954 (caymanchem, 28279) in DMSO, SRI-41315 [31] was a kind gift from the Cystic Fibrosis Foundation (was dissolve in DMSO), Apidaecin IB (Anaspec, AS-62044) in DDW, N-Oxalylglycine (caymanchem, 13944) in DMSO, Escin (Santa Cruz, SC-221596) in methanol, Ataluren (Caymanchem, 16758) in DMSO and Erythromycin (Caymanchem, 16486) in ethanol.

## Immunofluorescence

Cells were grown on 13 mm round coverslips and then fixed for 20 min in PBS containing 4% paraformaldehyde. After 3 washes with PBS, the fixed cells were permeabilized with 0.1% Triton X-100 for 10 min and blocked with bovine serum albumin for 1 h. Subsequently, cells were incubated at room temperature with rabbit anti-active β-catenin (1:250) and Alexa fluor 488 anti-rabbit (1:500) for 60 and 45 min, respectively; 4′,6-Diamidino-2-phenylindole (DAPI, Sigma 10 μg/ml) was used to stain cell nuclei. An independent script was used to quantify the RGB intensity of nuclear β-catenin.

## Western blot analysis

Cells were washed with PBS and solubilized in lysis buffer (50 mM Tris (pH 7.5), 100 mM NaCl, 1% Triton X-100, 2 mM EDTA) containing a protease inhibitor cocktail (Sigma). Full-length APC was detected by solubilizing CRC cell lines in 6 M urea lysis buffer (50 mM Tris (pH 7.5), 120 mM NaCl, 1% NP-40, 1 mM EDTA) containing protease inhibitor cocktail. Extracts were clarified by centrifugation at 12,000×g for 15 min at 4˚C. Following SDS poly-acrylamide gel electrophoresis (SDS-PAGE), proteins were transferred to nitrocellulose membranes and blocked with 5% low-fat milk. The membranes were then incubated with specific primary antibodies, washed with PBS containing 0.001% Tween-20 (PBST), and incubated with the appropriate horseradish peroxidase-conjugated secondary antibody. After washing in PBST, membranes were subjected to enhanced chemiluminescence detection analysis. Band intensities were quantified by Fusion-Capt analysis software.

**Table 1. qPCR primers.**

| | Fw primer | Rv primer |
|---|---|---|
| APC | 5′ GCTCTATGAAAGGCTGCATGAG 3′ | 5′ TCACACTTCCAACTTCTCGC 3′ |
| ACTIN | 5′ CCTGGCACCCAGCACAAT 3′ | 5′ GGGCCGGACTCGTCATACT 3′ |

### RNA isolation and RT-qPCR analysis

Total RNA was isolated from the cultured cells using TRI reagent (Bio-lab) and an RNA extraction kit (ZYMO) according to the manufacturer's protocol. Total RNA (1 μg) was reverse transcribed using the iScript cDNA Synthesis Kit (Bio-Rad) according to the manufacturer's instructions. Real-time PCR was performed using the CFX Connect Real-Time PCR Detection System (Bio-Rad) using a SYBR Green Master mix (PCR Biosystems). Actin was used as a housekeeping control. All reactions were in triplicates. The primers for the amplification of the specific cDNA sequences were (Table 1).

### Cell viability assay

PrestoBlue viability reagent (Thermo Fisher, A13261) was used according to the manufacturer's protocol. Measurement of absorbance at 570 and 600 nm, using Epoch microplate spectrophotometer (BioTek). All the treatments were in triplicates.

### Statistical analysis

Data were analyzed using GraphPad Prism software and are presented as the mean with standard deviation. Analysis of variance (ANOVA) was performed when appropriate to assess the significance of variations using Tukey's multiple comparisons. *P* values are as indicated.

## Results

### Phosphorylation of the mTOR substrate 4EBP-1 facilitates the antibiotic-mediated readthrough response to serum starvation in cancerous cell lines

We have previously shown that stress induced by serum starvation increases antibiotic-mediated nonsense mutation readthrough using both a reporter-based cellular system and the endogenous APC gene product in the CRC cell line Colo320 [29]. To understand the physiological mechanisms underlining this effect, we tested whether serum starvation may enhance antibiotic-mediated readthrough in additional cell lines (harboring different endogenous APC nonsense mutations). Cells were incubated for 24 h in a medium containing 10% or 1% serum supplemented with 1.5 mg/ml G418. The results show that antibiotic-mediated nonsense mutation readthrough was increased in 3 (SW837, SW620, and DU4475) out of the 4 tested cell lines when the serum concentration was reduced (Fig 1A). The APC tumor suppressor is an essential component of a cytoplasmic protein complex that targets β-catenin for destruction [32]. Thus, the functionality, at least partly, of the restored APC was demonstrated by its ability to reduce the levels of active β-catenin (Fig 1A; middle blot). Unlike the results obtained using the Colo320 cells [29], despite APC enhanced restoration, active β-catenin levels were not further reduced following serum depletion. Different studies have shown that the levels of β-catenin do not directly correlate to the levels of full-length APC (for example, as shown in [33–37]). This phenomenon is thought to represent a threshold of β-catenin expression. Importantly, it was shown that the number of β-catenin binding sites in the APC protein affects its ability to decrease β-catenin levels, though not in an absolute correlation [38]. Our results using the Colo320 cells in which, uniquely, the APC has no β-catenin binding sites [39,40]

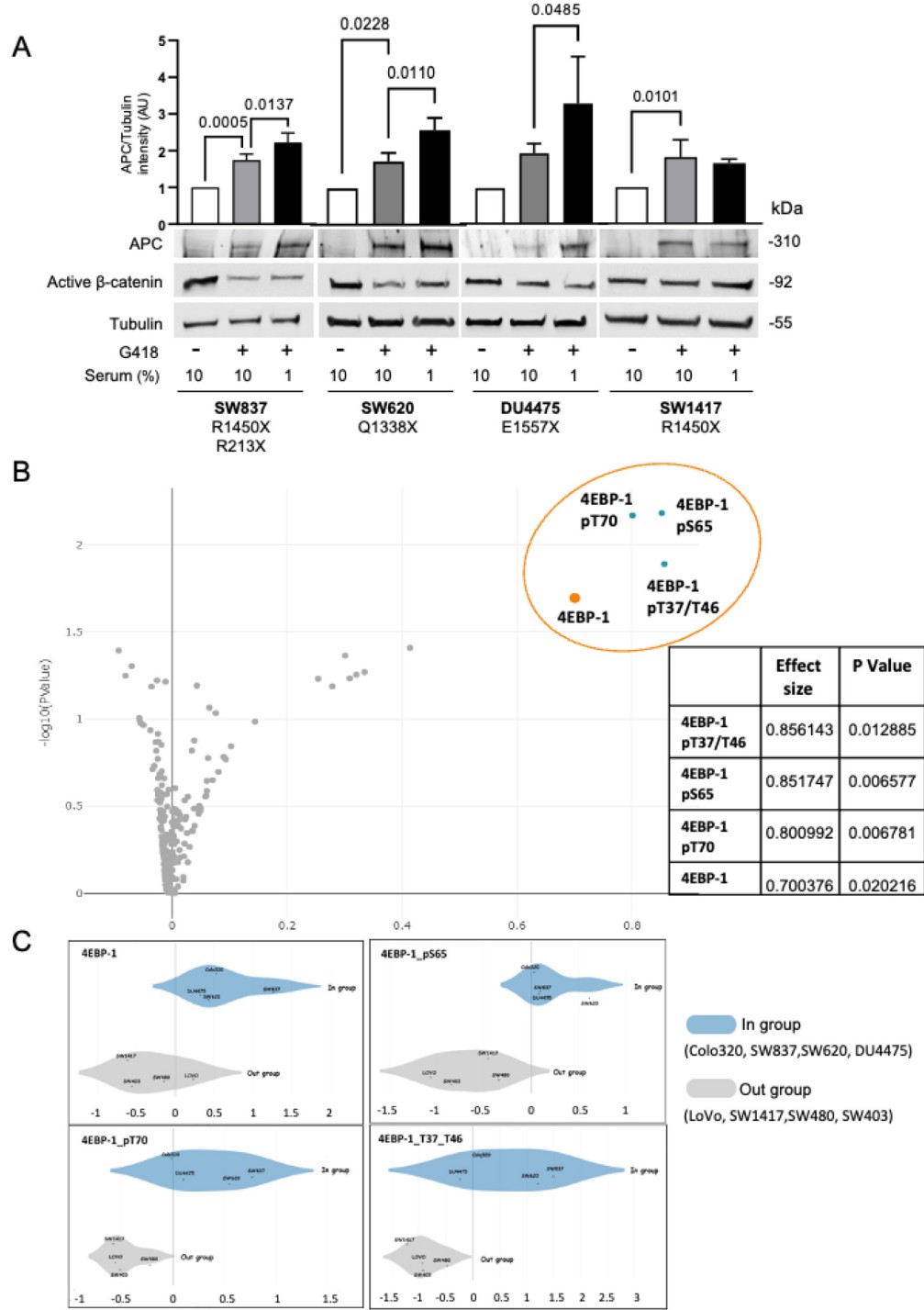

**Fig 1. Phosphorylation of the mTOR substrate 4EBP-1 facilitates the antibiotic-mediated readthrough response to serum starvation in cancerous cell lines.** **(A)** Colon carcinoma SW837, SW620, SW1417, and breast cancer DU4475 cell lines were supplemented with 10% or 1% serum (serum starvation) as indicated and treated with 1.5 mg/ml G418 for 24 h followed by WB analysis using antibodies specific for APC, active β-catenin, and tubulin. The specific nonsense mutations in APC in each cell line are indicated. The graphs represent APC/tubulin band intensities (arbitrary units) calculated by Fusion-Capt analysis software. Bars represent mean values ± SD of 2–5 independent experiments. One-way ANOVA tests were conducted for SW837: $P < 0.0001$, SW620: $P = 0.0006$, SW117: $P = 0.0085$, and DU4475: $P = 0.0027$. Tukey's multiple comparisons scores are shown. **(B)** Volcano plot showing a two-class comparison of protein expression levels between cell lines in which serum starvation increased APC readthrough (Colo320, SW620, SW837, and DU4475

—*in-group*) and non-responsive cell lines (SW403, SW480, SW1417, and LOVO—*out-group*) was conducted using the Broad Institute DepMap web portal (all the proteins available for these cell lines, in this database). **(C)** Two-class comparison of protein expression of 4EBP-1 and phosphorylated forms between cell lines in which serum starvation increased APC readthrough (Colo320, SW620, SW837, and DU4475) and non-responsive cells (SW403, SW480, SW1417, and LOVO) was conducted using the Broad Institute DepMap web portal (https://depmap.org/portal/download/). The data underlying the graphs in the figure can be found in S1 Data. ANOVA, analysis of variance; 4EBP-1, 4E (eIF4E)-binding protein 1; APC, adenomatous polyposis coli; mTOR, mammalian target of the rapamycin; WB, western blot.

show the strongest effect on β-catenin expression levels [29]. For example, SW837 and SW620 still retain β-catenin binding sites [40–42], and thus restored full-length APC may not directly affect the expression of β-catenin. Thus, in most experiments (Figs 2–5 and S2B), the Colo320 cells were used. In addition, other mechanisms, such as neddylation [43], GSK3β mTOR-dependent phosphorylation [44], and microRNAs' expression [45], that have been shown to be involved in regulating β-catenin levels may be influenced by the readthrough-inducing drugs and restrict the effects on β-catenin.

The antibiotic-mediated nonsense mutation readthrough specificity was confirmed by comparing CRC cell lines without an APC nonsense mutation to the Colo320 cells (S1A Fig). Indeed, HCT116, which encodes for a full-length APC protein, and SW48, which carries an APC missense mutation, were not affected by the G418 treatment, and active β-catenin protein levels were unchanged (S1A Fig). Moreover, neither G418 nor serum deprivation had any effect on the truncated APC (Tr-APC) protein expressed in the different cell lines tested (S1B and S1C Fig).

To understand why reduced serum levels enhanced antibiotic-mediated readthrough only in certain cell lines, a two-class comparison of protein expression levels using the Broad Institute DepMap web portal was performed using serum depletion responsive cells (Colo320, SW837, SW620, and DU4475) ([29] and Fig 1A) compared to the non-responsive cells (SW1417, SW403, LoVo, SW480) ([29] and Fig 1A). The analysis results indicated statistically significant differences for 8 out of a total of 214 proteins, with the greatest differences noted for 4E (eIF4E)-binding protein 1 (4EBP-1) and its phosphorylated forms (Fig 1B). 4EBP-1, a member of a family of translation repressor proteins, is a known direct substrate of the mammalian target of the rapamycin (mTOR) signaling pathway [46]. The mTOR pathway controls, among other things, the cell response to nutrients via translation regulation [33]. Activated mTOR complex1 (mTORC1) signaling leads to the phosphorylation of several downstream components, including S6K1 and 4EBP-1. When stimulated, mTORC1 mediates the phosphorylation of 4EBP-1, which in turn releases eukaryotic translation initiation factor 4E (eIF4E) to initiate 5′cap-dependent protein synthesis [47]. In our analysis, 4EBP-1 and its 3 phosphorylated forms were highly and abundant in the cells that demonstrated increased antibiotic induced-readthrough of nonsense mutations following serum depletion (Fig 1C).

## mTOR inhibition increases antibiotic-induced nonsense mutation readthrough

mTOR is a highly conserved serine/threonine kinase complex that controls cell growth and metabolism [48]. By regulating protein synthesis, mTORC1 controls the levels of available cellular energy substrates and maintains the availability of the amino acid pool. Dysregulation of the mTOR pathway leads to aberrant translation and various pathological conditions [49]. When amino acids are scarce (as in the case of serum starvation), mTOR is inactivated to reduce protein production [50]. To assess the involvement of the mTOR cascade in

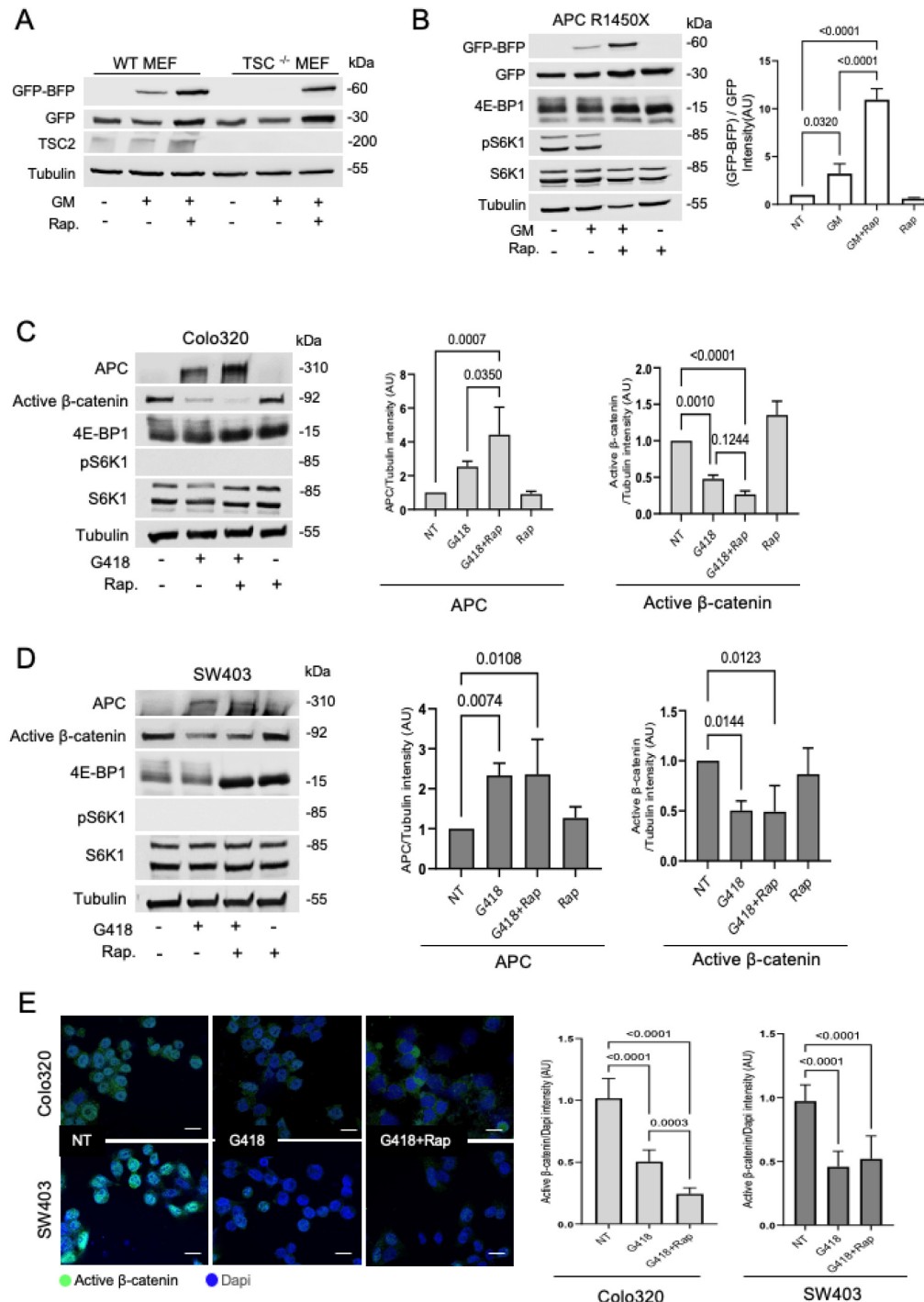

**Fig 2. mTOR inhibition increases antibiotic-induced nonsense mutation readthrough. (A)** WT MEF and TSC -/- MEF cells were seeded in a 24-well plate before transfection and transiently transfected with the GFP-Stop-BFP construct (S1278X). The cells were treated with 1.5 mg/ml Gentamicin (GM) or 1 μm Rapamycin (Rap) for 24 h followed by WB analysis with the indicated antibodies. **(B)** The APC 1450X reporter cell line was treated for 24 h with 500 μg/ml GM or 1 μm Rap followed by WB. The graphs show the relative GFP-BFP band intensity (normalized to GFP band intensity). Bars represent the mean values ± SD from 5 independent experiments. $P < 0.0001$. **(C, D)** Colo320 (C) and SW403 (D) were treated for 24 h with 500 μg/ml G418 or 1 μm Rap followed by WB. Graphs represent the intensities of the APC/tubulin or active β-catenin/tubulin bands (in arbitrary units), calculated by the Fusion-Capt analysis software. The bars represent the mean values ± SD from 4–6 independent experiments. Colo320: APC $P = 0.0005$, active β-catenin $P < 0.0001$, SW403: APC $P = 0.0031$, active β-catenin $P = 0.0048$. **(E)** Colo320 (1 experiment) and SW403 (2 independent

experiments) were treated with 1.5 mg/ml G418 and 25 μm Rap for 24 h. The cells were then fixed and visualized by confocal microscopy. The graph represents the active β-catenin intensity (green) normalized to DAPI (blue) in antibiotic-treated cells. An independent script was used to quantify the RGB intensity of nuclear active β-catenin in 10 independent fields of each sample. $P < 0.0001$. Tukey's multiple comparison scores are shown. The scale bars represent 20 μm. The data underlying the graphs in the figure can be found in S1 Data. APC, adenomatous polyposis coli; MEF, mouse embryo fibroblast; mTOR, mammalian target of the rapamycin; WB, western blot.

antibiotic-mediated nonsense mutation readthrough, the effect of the tuberous sclerosis complex 1/2 genes (TSC1/2) that negatively regulates mTOR signaling [51] was assessed. Immortalized TSC1/2$^{-/-}$ mouse embryo fibroblasts (MEFs), which have a constitutively high mTOR activity [30,51], were transfected with a reporter plasmid (APC R1450X) expressing a readthrough-sensitive chimeric GFP-BFP cassette that harbors a specific stop codon (and surrounding sequences) between the GFP and BFP open reading frames [18]. The expression levels of the upstream GFP protein serves as a control for total translation initiation of the chimeric protein expression, while the levels of the GFP-BFP chimeric protein reflect the degree of stop codon readthrough activity. Both products can be measured by western blot analysis. The effect of Gentamycin (GM) on readthrough in TSC1/2$^{-/-}$ was compared to that in wild-type (WT) MEFs. Interestingly, antibiotic treatment of the TSC1/2$^{-/-}$ cells did not induce readthrough (Fig 2A), and thus, no GFP-BFP fused protein was detected, although the WT cells did express the GFP-BFP fused protein following treatment. These results suggest that the mTOR pathway may be involved in antibiotic-mediated nonsense mutation readthrough. This notion is supported by the observation that treating TSC1/2$^{-/-}$ cells with the mTOR inhibitor Rapamycin increases the ability of the antibiotic to induce stop codon readthrough (Fig 2A). To explore the involvement of the mTOR pathway in antibiotic-mediated nonsense mutation readthrough, we tested 2 known inhibitors of the mTOR cascade: Torin-1 (S2 Fig) and Rapamycin (Fig 2). Torin-1 is a synthetic mTOR inhibitor that blocks ATP binding to mTOR and thus inactivates both mTORC1 and mTORC2 [52], whereas Rapamycin is a macrolide known to selectively target mTORC1 and inhibit cap-dependent mRNA translation [53]. To decrease toxicity-related stress, the antibiotic concentration was reduced to 500 μg/ml (cell survival of 60% to 100%; S3 Fig). Treating cells stably expressing the GFP-BFP reporter plasmid (APC R1450X) with either Torin-1 or Rapamycin in the presence of GM, enhanced antibiotic-mediated PTC readthrough (Figs 2B and S2A, respectively). The effect of mTOR inhibition was next examined in the CRC cell line Colo320, where serum starvation was shown to enhance G418-mediated readthrough [29]. Treated Colo320 cells demonstrated relatively high levels of APC restoration following antibiotic treatment, and importantly, APC readthrough was further increased in response to mTOR inhibition (Torin-1; S2B Fig or Rapamycin; Fig 2C). The levels of active β-catenin were reduced accordingly indicating the functionality of the restored APC protein. Resembling the serum starvation effect, SW403 cells did not respond to mTOR inhibition, and the APC restoration levels were similar to those induced by the antibiotic alone (Torin-1; S2C Fig or Rapamycin; Fig 2D). This correlation between the response to mTOR inhibition and serum starvation was also observed in other cell lines (S4 Fig). Fig 2E demonstrates that, as expected, Rapamycin addition impeded the nuclear translocation of β-catenin only in Colo320 cells. Interestingly, although mutated APC transcripts are relatively stable, as they often escape NMD, a slight increase in APC mRNA levels was observed in treated Colo320 cells compared to SW403 cells that were unaffected by readthrough or readthrough enhancement (S5 Fig).

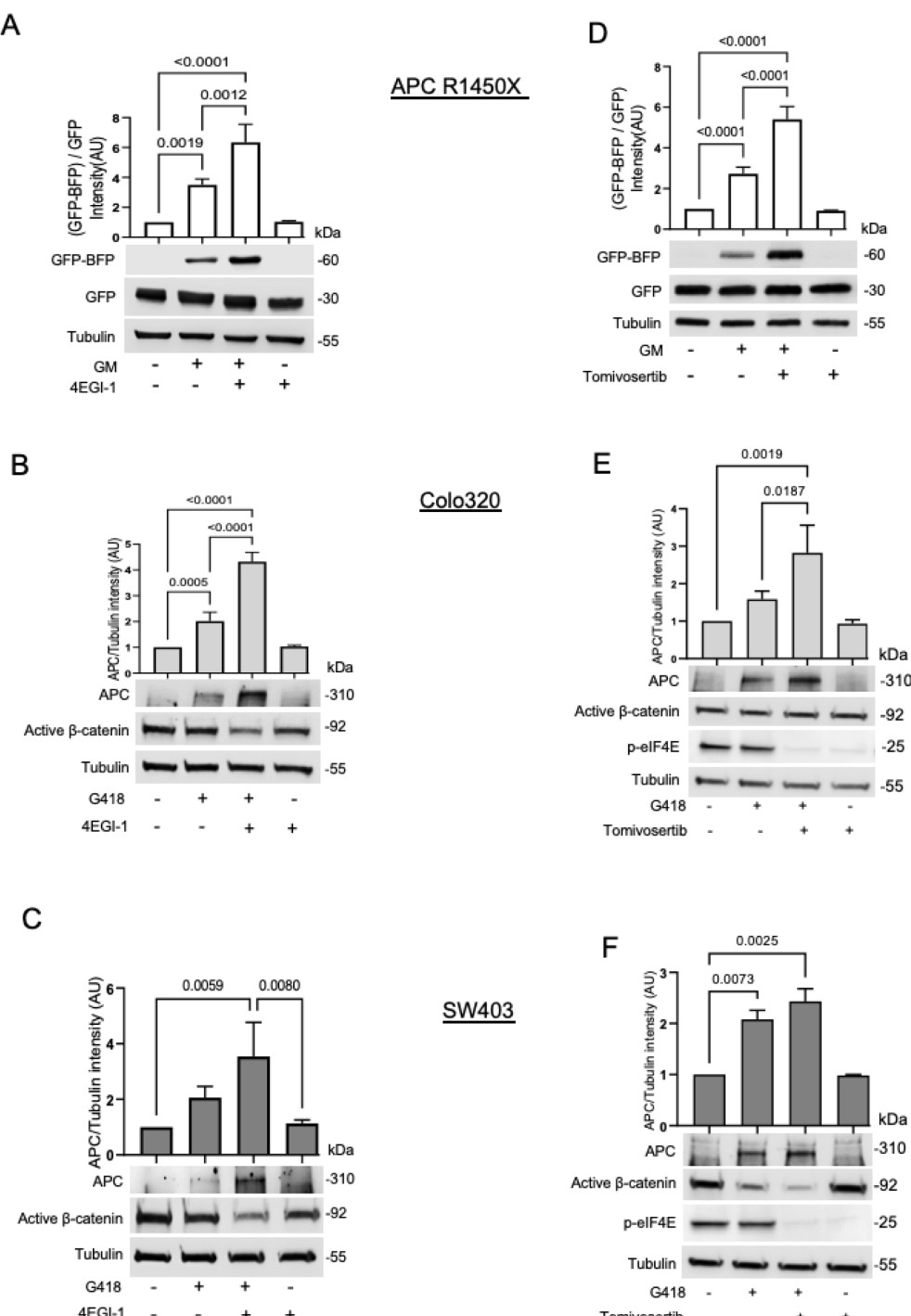

**Fig 3. Reduced cap-dependent translation initiation increases antibiotic-mediated nonsense mutation readthrough.**
(**A**) The APC R1450X reporter cell line was treated for 24 h with 500 µg/ml GM and/or 50 µm 4EGI-1 (eIF4E/eIF4G Interaction Inhibitor). The graphs represent the relative GFP-BFP band intensity (normalized to GFP band intensity). Bars represent the mean values ± SD from 3 independent experiments. $P = 0.0004$. (**B, C**) Colo320 (B) and SW403 (C) cell lines were treated for 24 h with 500 µg/ml G418 or 50 µm 4EGI-1. The bars represent the mean values ± SD from 3–4 independent experiments. Colo320: $P < 0.0001$; SW403: $P = 0.0047$, Tukey's multiple comparison scores are shown. (**D**) The APC R1450X reporter cell line was treated for 24 h with 500 µg/ml GM and/or 30 µm Tomivosertib (MNK Inhibitor). The graphs represent the relative GFP-BFP band intensity (normalized to GFP band intensity). Bars represent the mean values ± SD from 3–4 independent experiments. $P = 0.0004$. (**E, F**) Colo320 (E) and SW403 (F) cell

lines were treated for 24 h with 500 μg/ml G418 or 30 μm Tomivosertib. The bars represent the mean values ± SD from 2–3 independent experiments. Colo320: $P$ = 0.0011; SW403: $P$ = 0.0014. Tukey's multiple comparison scores are shown. The data underlying the graphs in the figure can be found in S1 Data. APC, adenomatous polyposis coli; elF4E, eukaryotic translation initiation factor 4E; eIF4G, eukaryotic translation initiation factor 4G; GM, gentamicin.

### Reduced cap-dependent translation initiation increases antibiotic-mediated nonsense mutation readthrough

Canonical translation initiation is mediated by ribosome binding to the 7-methyl-GTP group at the 5′ termini of eukaryotic mRNAs, termed 5′ cap structure. eIF4E plays a crucial role in 5′ cap-dependent translation initiation as it directly binds to the 5′ cap structure [54]. MAPK-interacting kinase (MNK) phosphorylates eIF4E at Ser209, using eukaryotic translation initiation factor 4G (elF4G) as a docking site [54–56]. The phosphorylation of eIF4E increases its affinity to the 5′ cap of structure and potentially facilitates its entry into the initiation complexes [57]. Together with eIF4G and eIF4A, eIF4E generates the eIF4F complex, which is fundamental to successful 5′cap-dependent translation initiation. 4EBP-1, the direct target of the mTORC1 pathway, serves as a translation inhibitor in its un-phosphorylated form as it binds to the 5′cap-binding protein eIF4E [58], preventing its association with eIF4G. The small-molecule inhibitor 4EGI-1 stabilizes the eIF4E/4EBP-1 bond, which hinders the elF4E-elF4G interaction, leading to translation initiation inhibition [59]. Our data demonstrate that 4EGI-1 enhances antibiotic-induced nonsense mutation readthrough to generate both the GFP-BFP fused protein in the APC R1450X reporter cell line (Fig 3A) and the endogenous APC protein in Colo320 CRC cells (Fig 3B). Interestingly, similar results were observed in the SW403 CRC cells (Fig 3C), although their readthrough levels are not affected by serum starvation. This result may indicate that directly inhibiting cap-dependent translation initiation regardless of serum levels is sufficient for enhancing antibiotics-induced nonsense mutation readthrough. We thus tested an additional inhibitor that targets translation initiation. We used Tomivosertib, a potent and highly selective dual MNK1/2 inhibitor [60], to impede eIF4E phosphorylation and inhibit 5′ cap-dependent translation initiation. Similarly to 4EGI-1, MNK inhibition enhances antibiotic-induced PTC readthrough of the GFP-BFP fusion protein in the APC R1450X reporter cell line (Fig 3D). The expression levels of full-length endogenous APC in both the Colo320 (Fig 3E) and SW403 CRC cell lines (Fig 3F) were increased in response to Tomivosertib (when combined with G418). The effects of APC restoration on active β-catenin levels are shown in S6 Fig (4EGI-1; A-B, Tomivosertib; C-D).

Ribosomal protein S6 kinase beta-1 (S6K1) phosphorylation is also a known stimulus of protein synthesis [61]; however, our results indicate that inhibiting S6K1 (using the S6K1 inhibitor (PF-4708671)) did not affect antibiotic-mediated nonsense suppression in the APC R1450X reporter cell line (S7 Fig). Interestingly, since Colo320 and SW403 do not express p-S6K1 (Figs 2C and 2D and S2B and S2C), we conclude that S6K1 is not involved in the effects of mTOR inhibition on antibiotic-mediated nonsense mutation readthrough.

### Targeting different stages of the protein synthesis process enhances aminoglycoside-mediated nonsense mutation readthrough

During the elongation step of mRNA translation, aminoacyl-charged tRNAs are recruited into the aminoacyl (A) site of the ribosome by eukaryotic elongation factor 1A (eEF1A). Next, following peptide bond generation, the eukaryotic elongation factor 2 (eEF2) GTPase, which is regulated by the highly specific eEF2 protein kinase (eEF2K) [62], enables the translocation of the ribosome to the next codon to allow the following decoding event [63]. Inactivating eEF2K enhances eEF2 activity leading to accelerated elongation that is usually accompanied by

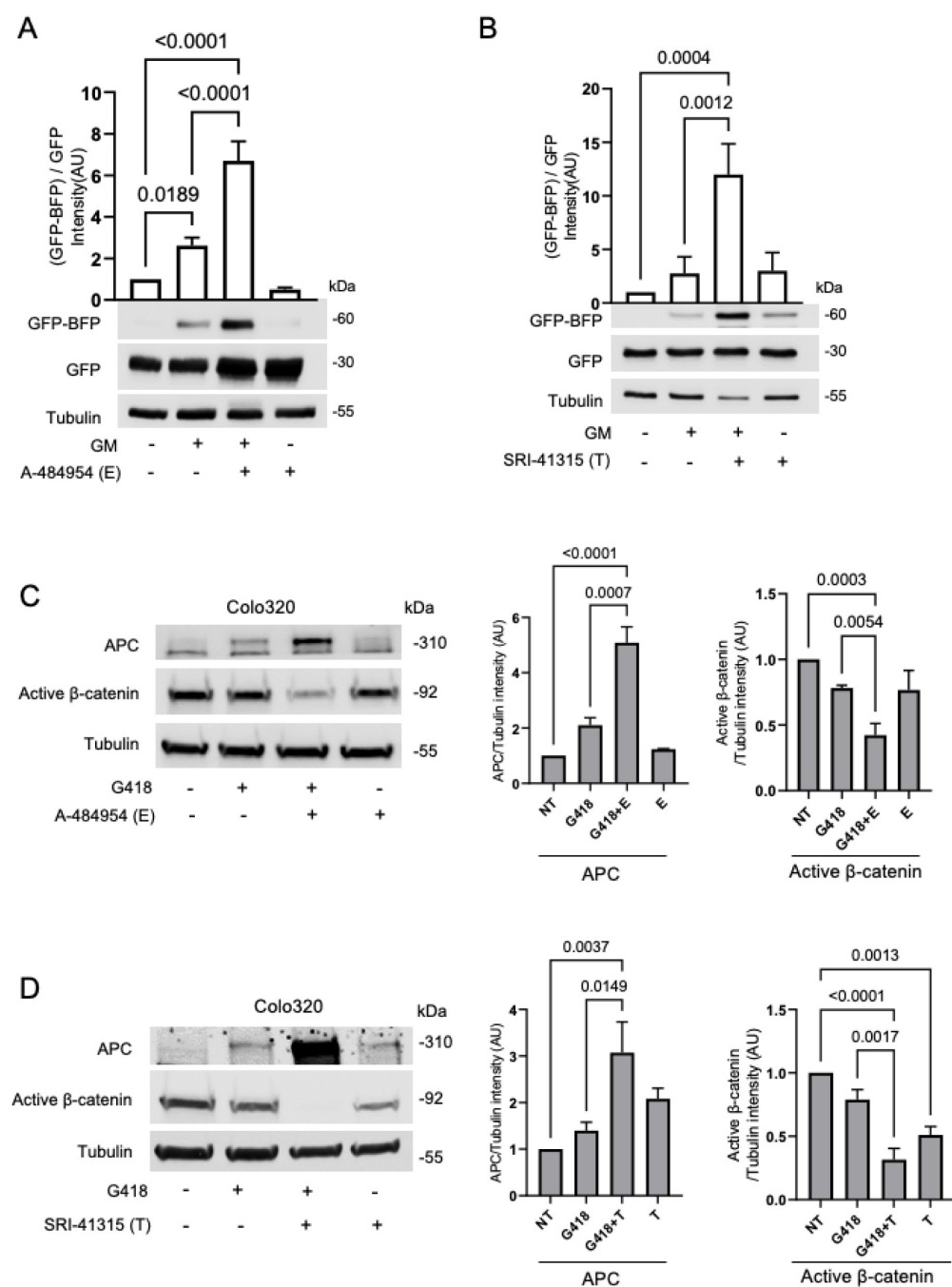

**Fig 4. Targeting different stages of the protein synthesis process enhances aminoglycosides-mediated nonsense mutation readthrough. (A)** The APC R1450X reporter cell line was treated for 24 h with 500 µg/ml of Gentamicin (GM) and/or 100 µm A-484954 (eEF2K inhibitor, E). The graphs represent the relative GFP-BFP band intensity (normalized to GFP band intensity). Bars represent the mean values ± SD from 3 independent experiments. *P* = 0.0004. **(B)** The APC R1450X reporter cell line was treated for 24 h with 500 µg/ml GM and/or 5 µm SRI-41315 (eRF1 inhibitor, T). The graphs represent the relative GFP-BFP band intensity (normalized to GFP band intensity). Bars represent the mean values ± SD from 3 independent experiments. *P* = 0.0004. **(C)** Colo320 cell line was treated for 24 h with 500 µg/ml G418 and/or 100 µm A-484954. The bars represent the mean values ± SD from 3 independent experiments. *P* < 0.0001. **(D)** Colo320 cell line was treated for 24 h with 500 µg/ml G418 and/or 5 µm SRI-41315. The bars represent the mean values ± SD from 3–4 independent experiments. *P* < 0.0001. The data underlying the graphs in the figure can be found in S1 Data. APC, adenomatous polyposis coli; eEF2K, eukaryotic elongation factor 2 kinase; eRF1, translation termination factor 1; GM, gentamicin.

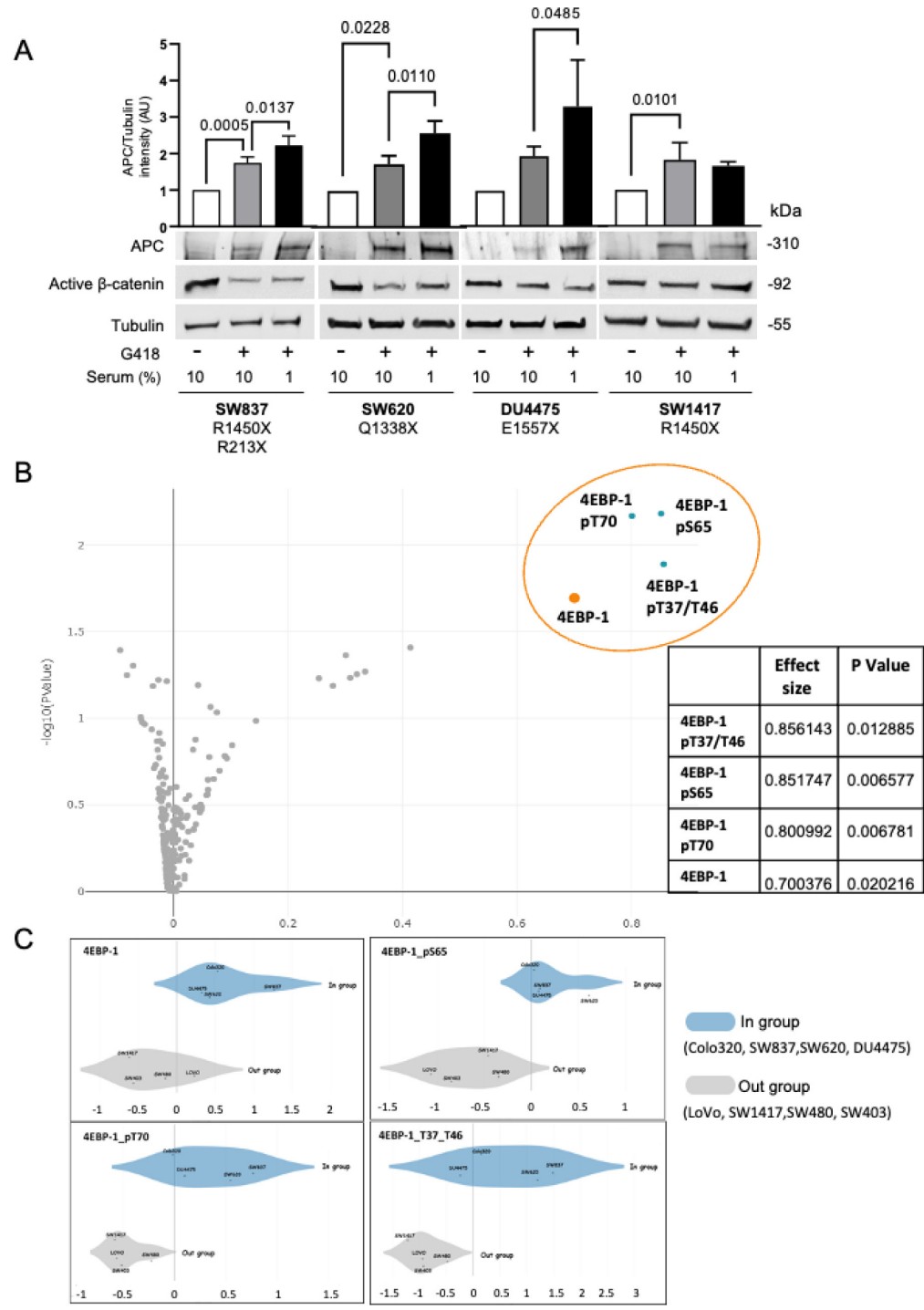

**Fig 5. Targeting different stages of the protein synthesis process enhances readthrough mediated by non-aminoglycosides. (A)** The APC R1450X reporter cell line was treated for 24 h with 10 μm Escin or 200 μm Ataluren and 50 μm 4EGI-1, 100 μm A-484954 or 5 μm SRI-41315. **(B)** The APC R1450X reporter cell line (upper blot) and Colo320 cell line (lower blot) were treated for 24 h with 300 μg/ml Erythromycin and 50 μm 4EGI-1, 100 μm A-484954 or 5 μm SRI-41315. APC, adenomatous polyposis coli.

impaired translational fidelity [62,63]. To test the effect of elongation rate on readthrough activity, we used A-484954, a highly selective eEF2K inhibitor [64]. As shown in Fig 4, eEF2K inhibition enhances aminoglycoside-induced nonsense mutation readthrough of both the GFP-BFP fusion protein in the APC R1450X reporter cell line (Fig 4A) and endogenous APC in Colo320 (Fig 4C). This result indicates that accelerated elongation can increase aminoglycoside-mediated nonsense mutation readthrough.

Next, we examined whether targeting translation termination may also influence antibiotic-induced PTC. Translational readthrough efficiency depends on competition between stop codon recognition by eukaryotic translation termination factor 1 (eRF1) and decoding of the stop codon by a near-cognate tRNA [65]. SRI-41315 induces translational readthrough by depleting eRF1 levels through a proteasome-mediated degradation pathway [31]. We demonstrate that SRI-41315 enhances antibiotic-induced nonsense mutation readthrough of both the GFP-BFP fused protein in the APC R1450X reporter cell line (Fig 4B) and endogenous APC in Colo320 (Fig 4D). SRI-41315 treatment alone induced low levels of nonsense mutation readthrough in both experiments (Fig 4B and 4D).

Apidaecin, an 18-amino acid proline-rich antimicrobial peptide from honeybees, was recently shown to have RF1- and RF2-inhibiting activity in bacteria [66–69]. An additional eRF1 inhibitor is NOG (dimethyloxalylglycine; DMOG), which reduces the hydroxylation of newly synthesized eRF1 [70]. Inhibition of eRF1 regular activity by Apidaecin or NOG-induced antibiotic-induced nonsense mutation readthrough of the GFP-BFP chimeric protein in APC R1450X reporter cell line (S8 Fig).

## Targeting different stages of the protein synthesis process enhances readthrough mediated by non-aminoglycosides

Taken together, we conclude that the modulation of different stages of the translation machinery affects the general mechanism of antibiotic-mediated nonsense suppression. To determine whether this is specific to aminoglycosides, we tested additional non-aminoglycoside inducers in both our reporter system and in Colo320 CRC cells. We used Erythromycin, a macrolide that was shown to induce nonsense suppression in several systems, including a clinical trial [18,71], Escin, a natural herbal product that was shown to induce readthrough [72], and Ataluren (PTC124), which is a benzoic acid derivative that was shown to induce readthrough in various syndromes and model systems. Ataluren was recently approved to treat DMD patients carrying nonsense mutations (reviewed in [73]). As shown in Fig 5, the effects of all readthrough-enhancing agents were enhanced by modifying the initiation, elongation, and termination of the protein synthesis process. These results show that manipulation of different translation steps serves as a general approach to enhance nonsense suppression and can be applied to induce readthrough mediated by both aminoglycosides and non-aminoglycosides compounds.

## Discussion

Single nucleotide substitutions in gene coding regions can change a sense codon into a nonsense or (i.e., PTC). The resultant PTC-containing mRNA may then be degraded by the nonsense-mediated mRNA decay cellular surveillance pathways or translated into a truncated, mostly nonfunctional protein. In either case, early termination codon may lead to a wide range of human genetic disorders. One such example is nonsense mutations in the APC gene in both hereditary and sporadic CRC. Generally, suppression of nonsense mutations is a viable therapeutic strategy that has been studied extensively. Possible approaches to overcome the disease-causing phenotype of these mutations include antisense oligonucleotides, suppressor

tRNAs that can read PTCs, RNA editing, or CRISPR technology [4]. However, the most commonly studied methods for inducing PTC readthrough are by using aminoglycosides [8,74–76], macrolides [18], synthetic aminoglycosides such as ELX-02 (NB124) [77,78], and oxadiazole derivates such as Ataluren (PTC124) [79,80]. Unfortunately, these compounds have limitations, including high toxicity and adverse side effects [15]. As a result, strategies that reduce dosage or decrease treatment frequency while increasing treatment efficacy are greatly needed. Although induced nonsense mutation suppression has been studied and tested for decades, a better understanding of the molecular mechanisms that control this important potential therapeutic strategy is still essential for developing more potent treatment approaches.

Our initial results suggested the involvement of cellular nutrients levels and the mTOR substrate 4EBP-1 in antibiotic-mediated nonsense suppression. 4EBP-1, along with several of its phosphorylated forms, were shown to be differentially expressed in cells that either responded or did not respond to serum deprivation-induced nonsense mutation readthrough (Fig 1 and [29]). The eIF4E/eIF4G translation initiation sub-complex is active when mTOR-dependent phosphorylation of 4EBP-1 occurs [58]. The small molecule 4EGI-1, which inhibits 5′ cap-dependent translation initiation by preventing eIF4E/eIF4G association [59,81], efficiently increases antibiotic-mediated readthrough. 4EGI-1 is of particular interest since it has anti-tumorigenic activity and reduces the growth of human cancer xenografts in vivo [82]. The finding that antibiotic-mediated nonsense suppression is susceptible to translation initiation rate was further demonstrated by MNK inhibition that impedes eIF4E phosphorylation and inhibits translation initiation (Fig 3D–3F). Numerous MNK inhibitors (e.g., BAY1143269, Tomivosertib, CGP 57380, and ETC-206) have been tested in various clinical trials as a potential therapeutic strategy for cancer treatment [60]. Our results also demonstrate that mTOR inhibitors such as Torin-1 and Rapamycin, which lead to decreased 5′ cap-dependent translation initiation (by inhibiting the phosphorylating of 4EBP-1) augmented PTC readthrough as examined in a reporter fusion protein and the endogenous APC protein (Figs 2 and S2).

Here, we focused on studying the restoration of the tumor suppressor APC. Mutated APC transcripts are often NMD-resistant as most of the nonsense mutations occur in a hotspot within the last APC exon, and therefore, not recognized by the exon junction complex that induces NMD.

Aminoglycoside-induced readthrough requires binding at the decoding center of the eukaryotic ribosome and interference with the accurate recognition of the PTC by translation termination factors [83,84]. In this context, aminoglycoside-mediated readthrough has been shown to be enhanced by the depletion of eRF1, which induces translational pausing at PTCs (Fig 4) [31,77,85]. In addition, our results demonstrate that accelerating translation elongation also increased antibiotic-mediated readthrough activity (Fig 4A and 4C), indicating that affecting translation rates by manipulating different steps of the protein synthesis process significantly affects induced-nonsense suppression. Importantly, compounds that affect translation, such as Rapamycin, are in clinical use and may be repurposed or modified for treating diseases caused by PTCs in key genes.

Aminoglycosides exert their nonsense readthrough activity by binding at the eukaryotic ribosome's decoding center and reducing translation termination factors' ability to accurately recognize the PTC [83,84]. Other compounds such as the small molecules CDX5-1 [19] or the drug mefloquine [86], which do not induce readthrough when used as single agents, enhance aminoglycosides-mediated readthrough activity, possibly by targeting the translation machinery in a still unclear mechanism. As aminoglycosides induce readthrough by reducing ribosomal proofreading during translation, a combination with agents that affect translation may lead to the development of improved readthrough-inducing compounds.

Non-aminoglycosides induce nonsense suppression using a mechanism that may differ from that of aminoglycosides. Thus, the effects of translation modifications were also tested on induced readthrough, which was mediated by non-aminoglycoside compounds. Our results suggest that modulating the translation system affects readthrough in a general way and does not only rely on readthrough induced by aminoglycosides.

In conclusion, our findings indicate that nonsense suppression induced by various compounds can be improved by targeting different stages of the mRNA process by the translation machinery. Inhibition of initiation, which leads to a reduced number of ribosomes per mRNA molecule, increased elongation rate, or attenuation of termination, all augment the antibiotic-dependent misreading activity. These effects are specific to the stage and mechanism of translation as, for example, inhibition of S6K1 does not affect this process, and although 4EBP-1 inhibition and S6K activation are both downstream of mTORC1 activation and both promote protein synthesis, only decreasing translation initiation by activating 4EBP-1 (and not S6K) affects antibiotic-mediated nonsense mutation readthrough. Thus, further studies of these findings may improve our abilities to stimulate PTC readthrough for the treatment of genetic diseases caused by nonsense mutations.

## Supporting information

**S1 Fig. Induced APC nonsense mutation readthrough is specific and enhances the expression of the full-length protein. (A)** HCT116, SW48, and Colo320 cell lines were treated with 500 μg/ml G418 for 24 h followed by WB analysis using the indicated antibodies. **(B)** SW403 and LOVO cell lines were treated with 1.5 mg/ml G418 for 24 h followed by WB analysis using the indicated antibodies. FS = Frameshift. **(C)** Colo320 and SW480 cell lines were treated as in A, followed by WB analysis using antibodies specific for APC and tubulin.
(PPTX)

**S2 Fig. Torin-1 increases antibiotic-induced nonsense mutation readthrough. (A)** The APC R1450X reporter cell line was treated for 24 h with 500 μg/ml GM and/or 500 nM Torin-1 followed by WB. The graphs show the relative GFP-BFP band intensity (normalized to GFP band intensity). Bars represent the mean values ± SD from 4 independent experiments. $P < 0.0001$. **(B, C)** Colo320 (B) and SW403 (C) were treated for 24 h with 500 μg/ml G418 and/or 500 nM Torin-1 followed by WB analysis. Graphs represent the intensities of the APC/tubulin or active β-catenin/tubulin bands (arbitrary units), calculated by the Fusion-Capt analysis software. The bars represent the mean values ± SD from 2–4 independent experiments. Colo320: $P < 0.0001$, SW403: APC $P = 0.0023$, Active β-catenin $P < 0.0001$. Tukey's multiple comparisons scores are shown. The data underlying the graphs in the figure can be found in S1 Data.
(PPTX)

**S3 Fig. The effects of treatment on cell survival.** The APC-1450X reporter cell line, Colo320 and SW403 cell lines were treated for 24 h with 500 μg/ml GM (APC-1450X reporter cell line) or G418 (Colo320 and SW403), 500 nM Torin-1 or 1 μm Rap. PrestoBlue reagent was added to the wells and absorbance was measured after 3 h incubation at 570 and 600 nm. The bars represent the mean values ± SD from 3 independent experiments for each treatment, compared to untreated cells in each cell line. The data underlying the graphs in the figure can be found in S1 Data.
(PPTX)

**S4 Fig. APC restoration in additional CRC cell lines.** SW620 and SW837 were treated for 24 h (SW620) or 48 h (SW837) with 500 μg/ml G418 and/or 500 nM Torin-1 followed by WB

analysis for the indicated proteins.
(PPTX)

**S5 Fig. APC mRNA levels in the different treatments.** Colo320 and SW403 cell lines were treated for 24 h with 500 μg/ml G418 and/or 1 μm Rap. Total RNA was extracted from the treated samples, converted to cDNA, and subjected to RT-qPCR analysis. APC transcript levels were analyzed. The bars represent the mean values ± SD from 5 independent experiments. Two-way ANOVA ($P = 0.0012$) with Tukey's multiple comparisons test was applied—significant scores were depicted. The data underlying the graphs in the figure can be found in S1 Data.
(PPTX)

**S6 Fig. Reduced cap-dependent translation initiation increases antibiotic-mediated readthrough—the effect on active β-catenin. (A, B)** Colo320 (A) and SW403 (B) cell lines were treated for 24 h with 500 μg/ml G418 or 50 μm 4EGI-1. The bars represent the mean values ± SD from 3–4 independent experiments. Colo320: $P < 0.0001$; SW403: active β-catenin $P < 0.0001$. Tukey's multiple comparison scores are shown. **(C, D)** Colo320 (C) and SW403 (D) cell lines were treated for 24 h with 500 μg/ml G418 or 30 μm Tomivosertib. The bars represent the mean values ± SD from 3 independent experiments. Colo320: $P = 0.0073$; SW403: $P = 0.0058$. Tukey's multiple comparison scores are shown. The data underlying the graphs in the figure can be found in S1 Data.
(PPTX)

**S7 Fig. S6K1 phosphorylation is not involved in antibiotic-mediated nonsense mutation readthrough.** The APC 1450X reporter cell line was treated for 24 h with 500 μg/ml GM and 0.5, 1, 5, 10, or 20 μm PF-4708671 (S6K1 inhibitor) followed by WB analysis using the indicated antibodies. The graphs represent the relative GFP-BFP band intensity (normalized to GFP band intensity, with or without 20 μm PF-4708671), calculated by the Fusion-Capt analysis software. The bars represent the mean values ± SD from 4 independent experiments. The data underlying the graphs in the figure can be found in S1 Data.
(PPTX)

**S8 Fig. Additional termination inhibitors enhance antibiotic-mediated nonsense readthrough.** The APC 1450X reporter cell line was treated for 24 h with 500 μg/ml GM and 100 μg/ml Apidaecin or 500 μm N-Oxalylglycine (NOG). The bars represent the relative GFP-BFP band intensity (normalized to GFP band intensity) mean values ± SD from 3 independent experiments. $P < 0.0001$. The data underlying the graphs in the figure can be found in S1 Data.
(PPTX)

**S1 Data. Excel spreadsheet containing the underlying numerical data for Figs 1A, 2, 3, 4, S2, S3, S5, S6, S7, and S8.**
(XLSX)

**S1 Raw Images. The raw images for Figs 1A, 2A–2D, 3, 4, 5, S1, S2, S4, S7, and S8.**
(PDF)

## Acknowledgments

The authors thank Omri Kazelnik for technical advice and Julia Hofhuis for help with the figures. This work was performed in partial fulfillment of the requirements for the PhD degree of Amnon Wittenstein, Faculty of Medicine, Tel Aviv University, Israel.

## Author Contributions

**Conceptualization:** Amnon Wittenstein, Michal Caspi, Orna Elroy-Stein, Hagit Eldar-Finkelman, Rina Rosin-Arbesfeld.

**Data curation:** Amnon Wittenstein.

**Formal analysis:** Michal Caspi.

**Investigation:** Amnon Wittenstein, Rina Rosin-Arbesfeld.

**Methodology:** Amnon Wittenstein, Sven Thoms.

**Resources:** Ido Rippin.

**Supervision:** Rina Rosin-Arbesfeld.

**Writing – original draft:** Amnon Wittenstein, Michal Caspi, Sven Thoms, Rina Rosin-Arbesfeld.

**Writing – review & editing:** Rina Rosin-Arbesfeld.

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
