## [Editor Report · Decision Letter 0]

24 May 2023

Dear Dr. Rosin-Arbesfeld, 

Thank you for submitting your manuscript entitled "Nonsense mutation suppression is enhanced by targeting different stages of the protein translation process." for consideration as a Short Reports by PLOS Biology.

Your manuscript has now been evaluated by the PLOS Biology editorial staff and I am writing to let you know that we would like to send your submission out for external peer review.

Once your full submission is complete, your paper will undergo a series of checks in preparation for peer review. After your manuscript has passed the checks it will be sent out for review. To provide the metadata for your submission, please Login to Editorial Manager (https://www.editorialmanager.com/pbiology) within two working days, i.e. by May 26 2023 11:59PM.

Kind regards,

Paula Jauregui on behalf of 

Richard Hodge, 

Associate Editor

PLOS Biology

rhodge@plos.org

---

## [Decision Letter · Decision Letter 1]

4 Jul 2023

Dear Dr Rosin-Arbesfeld,

Thank you for your patience while your manuscript "Nonsense mutation suppression is enhanced by targeting different stages of the protein translation process." was peer-reviewed at PLOS Biology. Please accept my apologies for the delays that you have experienced during the peer review process. Your revised manuscript has now been evaluated by the PLOS Biology editors, an Academic Editor with relevant expertise, and by two of the original reviewers at Review Commons.

In light of the reviews, which you will find at the end of this email, we would like to invite you to revise the work to thoroughly address the reviewers' reports.

As you can see, whilst Reviewer #3 is now satisfied with the revised manuscript, Reviewer #1 raises some remaining concerns. Specifically, this includes the use of β-catenin as a marker for APC readthrough, the reporting/use of controls in several figures and a lack of data showing an increase in the efficiency of non-aminoglycoside readthrough compounds given the impact on different steps of translation. After discussions with Academic Editor, given that the concerns focus on data that was included in the revised manuscript after the initial review at Review Commons, we agree that these remaining comments should be addressed and that the inclusion of this data would help to generalize the findings and strengthen the overall message of the manuscript.

Given the extent of revision needed, we cannot make a decision about publication until we have seen the revised manuscript and your response to the reviewers' comments. Your revised manuscript is likely to be sent for further evaluation by all or a subset of the reviewers.

**IMPORTANT - SUBMITTING YOUR REVISION**

*Re-submission Checklist*

*Published Peer Review*

*PLOS Data Policy*

*Blot and Gel Data Policy*

Sincerely,

Richard

Richard Hodge, PhD

rhodge@plos.org

REVIEWS:

Reviewer #1: The manuscript proposed by Wittenstein et al. sets out to demonstrate that readthrough induced by antibiotics can be made more effective by acting on translation, in particular by inhibiting the initiation of cap-dependent translation, by increasing the activity of elongation or by inhibiting the translation termination step. Overall, the article is clear, the experiences are well described and organized in a coherent way. This article provides very interesting and useful data for the scientific community. I would only have a few points that would need to be addressed in order to strengthen the exposed data:

1) Figure 1A, the authors indicate that to test the functionality of APC restored by readthrough, they measure the expression level of beta-catenin. If we can clearly see that as soon as APC is increased, the level of beta-catenin decreases, it is however not proportional to the level of expression of APC. For example, in SW837 and SW620 cells at 1% serum, the level of beta-catenin is not lower than the 10% serum condition although the amount of APC produced is higher. The authors should comment on this point which is perhaps relative to a threshold of beta-catenin below which one cannot go?

2) Figure S3: the curve representation of the effect of different treatments on cell survival seems to me inappropriate because there is no continuum between the treatments. A representation in histogram would seem more appropriate.

3) Figure 4D, the authors indicate that the SRI-41315 readthrough molecule causes weak readthrough when used alone but it is not visible in the figure probably because the amount of protein loaded is lower, the Western-Blot should be redone by loading the same amount of protein in all lanes.

4) A point which seems essential to me is to show whether the impact on the different steps of translation also makes it possible to increase the efficiency of non-aminoglycoside readthrough compounds (Ataluren, Escin, 2,6-diaminopurine, etc…). These results would make it possible to generalize or not the impact of the different steps of translation on readthrough. The message of the article would then be much stronger.

Reviewer #3: The authors were very responsive to the reviewers' comments and have addressed all my concerns about this manuscript.

---

## [Decision Letter · Decision Letter 2]

15 Sep 2023

Dear Dr. Rosin-Arbesfeld,

Thank you for your patience while we considered your revised manuscript "Nonsense mutation suppression is enhanced by targeting different stages of the protein translation process." for publication as a Short Reports at PLOS Biology. This revised version of your manuscript has been evaluated by the PLOS Biology editors, the Academic Editor and one of the original reviewers.

Based on the reviews and our Academic Editor's assessment of your revision, we are likely to accept this manuscript for publication, provided you satisfactorily address the following data and other policy-related requests.

1. DATA POLICY:

A) Supplementary files (e.g., excel). Please ensure that all data files are uploaded as 'Supporting Information' and are invariably referred to (in the manuscript, figure legends, and the Description field when uploading your files) using the following format verbatim: S1 Data, S2 Data, etc. Multiple panels of a single or even several figures can be included as multiple sheets in one excel file that is saved using exactly the following convention: S1_Data.xlsx (using an underscore).

B) Deposition in a publicly available repository. Please also provide the accession code or a reviewer link so that we may view your data before publication.

Regardless of the method selected, please ensure that you provide the individual numerical values that underlie the summary data displayed in the following figure panels as they are essential for readers to assess your analysis and to reproduce it: Figures 1ABC, 2BCDE, 3ABCDEF, 4ABCD, and Supplementary Figures S2ABC, S3, S5, S6ABCD, S7, S8.

**Please also ensure that figure legends in your manuscript include information on where the underlying data can be found, and ensure your supplemental data file/s has a legend.**

We require the **original, uncropped and minimally adjusted images **supporting all blot and gel results reported in an article's figures or Supporting Information files. We will require these files before a manuscript can be accepted so please prepare and upload them now. We need this for Figures 1A, 2ABCD, 3ABCDEF, 4ABCD, 5AB and Supplementary Figures S1ABC, S2ABC, S4, S7, S8.

Please carefully read our guidelines for how to prepare and upload this data: https://journals.plos.org/plosbiology/s/figures#loc-blot-and-gel-reporting-requirements

3. Please add size bars to the microscopy pictures in Figure 2E.

We expect to receive your revised manuscript within two weeks.

*Published Peer Review History*

*Press*

Sincerely,

Paula Jauregui on behalf of

Richard Hodge,

Senior Editor,

rhodge@plos.org,

PLOS Biology

Reviewer remarks:

Reviewer #1: The authors answered my concerns and the article is in my opinion acceptable for publication.

---

## [Editor Report · Decision Letter 3]

29 Sep 2023

Dear Dr Rosin-Arbesfeld,

Thank you for the submission of your revised Research Article "Nonsense mutation suppression is enhanced by targeting different stages of the protein translation process." for publication in PLOS Biology. On behalf of my colleagues and the Academic Editor, Jeff Coller, I am pleased to say that we can accept your manuscript for publication, provided you address any remaining formatting and reporting issues. These will be detailed in an email you should receive within 2-3 business days from our colleagues in the journal operations team; no action is required from you until then. Please note that we will not be able to formally accept your manuscript and schedule it for publication until you have completed any requested changes.

PRESS

Kind regards, 

Richard

Richard Hodge, PhD

rhodge@plos.org

PLOS
